# The mTOR Signaling Pathway: Key Regulator and Therapeutic Target for Heart Disease

**DOI:** 10.3390/biomedicines13020397

**Published:** 2025-02-07

**Authors:** Jieyu Wang, Yuxuan Huang, Zhaoxia Wang, Jing Liu, Zhijian Liu, Jinfeng Yang, Zuping He

**Affiliations:** 1Department of Basic Medicine, School of Medicine, Hunan Normal University, Changsha 410013, China; 202370193524@hunnu.edu.cn (J.W.); 202320193586@hunnu.edu.cn (Y.H.); 202230193061@hunnu.edu.cn (Z.W.); liujing@hunnu.edu.cn (J.L.); 2The Key Laboratory of Model Animal and Stem Cell Biology in Hunan Province, Hunan Normal University, Changsha 410013, China; 3Department of Anesthesiology, Hunan Cancer Hospital/The Affiliated Cancer Hospital of Xiangya, School of Medicine, Central South University, Changsha 410013, China; alantempestliu@163.com

**Keywords:** mTOR, signaling pathway, heart disease, target

## Abstract

Heart disease, including myocardial infarction, heart failure, cardiac hypertrophy, and cardiomyopathy, remains a leading cause of mortality worldwide. The mammalian target of rapamycin (mTOR) is a centrally regulated kinase that governs key cellular processes, including growth, proliferation, metabolism, and survival. Notably, mTOR plays a pivotal role in cardiovascular health and disease, particularly in the onset and progression of cardiac conditions. In this review, we discuss mTOR’s structure and function as well as the regulatory mechanisms of its associated signaling pathways. We focus on the molecular mechanisms by which mTOR signaling regulates cardiac diseases and the potential of mTOR inhibitors and related regulatory drugs in preventing these conditions. We conclude that the mTOR signaling pathway is a promising therapeutic target for heart disease.

## 1. Introduction

For a long time, heart disease has remained a global public health issue that poses a serious threat to human health and causes a significant economic burden on society [1]. In recent years, despite medical advancements and social development, the age-standardized mortality rate of cardiovascular diseases has declined. However, this trend has plateaued, and the annual number of death continued to increase from 12.4 million to 19.8 million in 2022 [2]. This suggests that the health burden imposed by heart disease on human society is continuously intensifying, which highlights the severe impact of risk factors contributing to heart disease, e.g., aging, environmental factors, and air pollution [3]. Moreover, due to disparities in social development, residents of low- and middle-income countries face limited economic resources and relatively scarce access to medical care compared to those in developed nations. As a result, the health burden of heart disease in these populations is significantly heavier than in the developed countries of Western Europe. The pathogenesis of heart disease is multifactorial, involving both congenital factors, e.g., genetics and acquired factors, including diabetes, hypertension, infections, and unhealthy lifestyle habits [4,5,6]. Therefore, the onset of heart disease may be driven by one or more of these factors, and their combined effect often make its pathogenesis particularly complex. However, this complexity also presents opportunities for in-depth research and intervention, which could lead to treatment improvement.

Since discovering the mammalian target of rapamycin (mTOR), researchers have uncovered its extensive regulatory roles in various cellular functions. The mTOR signaling pathway plays a central role in key processes, e.g., cell growth, metabolism, proliferation, and autophagy [7]. Acting as a central regulatory factor and nutrient sensor, mTOR modulates cellular growth and metabolism by adjusting survival strategies and energy metabolism in response to changes in the external environment and energy availability [8,9]. When nutrients are abundant, mTOR is activated, which promotes cell growth, increases cell size, enhances cellular functions, and facilitates the synthesis of proteins, lipids, and nucleotides. Conversely, mTOR is inhibited under nutrient-deprived conditions, which triggers catabolic processes to supply the necessary energy and nutrients for cellular survival. In cellular proliferation, the mTOR signaling pathway regulates the cell cycle and it is a crucial component of oncogenic pathways. Active mTOR promotes the progression of cells through cell cycle checkpoints, thereby facilitating proliferation [10]. Inadequate nutrition or stress can inhibit mTOR and prevent cell proliferation. Excessive activation of the PI3K/AKT/mTOR signaling pathway stimulates uncontrolled cell proliferation and easily induces cancer [11,12]. The regulation of autophagy by mTOR is most closely linked to heart disease, particularly in the context of myocardial hypertrophy and ischemia–reperfusion injury [13,14]. mTOR complex 1 (mTORC1) plays a role in negatively regulating autophagy. When mTORC1 is inhibited (e.g., during RAPA treatment), the autophagic process is activated, which allows cells to degrade damaged or excess organelles and proteins, thereby recycling resources and clearing waste to maintain cellular homeostasis [15]. This feature is beneficial for preventing and alleviating organ damage and dysfunction caused by disease.

In recent years, researchers have further investigated the pathogenesis of heart disease and developed the prevention and treatment strategies of heart disease from the perspective of tissues and cells [16]. As studies on the mTOR signaling pathway have been deepened, there has been a growing understanding of its central regulatory role in cellular functions across an expanding range of fields. Inspired by these insights, researchers have increasingly linked mTOR with cardiovascular disease, progressively uncovering its critical role in the pathogenesis of heart disease and developing targeted therapeutic strategies aimed at mTOR [17,18,19].

## 2. The mTOR Pathway and Heart Disease

mTOR is an atypical serine/threonine kinase and a member of the phosphatidylinositol kinase-related kinase (PIKK) family [8,9,20]. It plays a pivotal role in regulating protein synthesis, cell growth, proliferation, autophagy, lysosome function, and cell metabolism. mTOR has also been implicated in various diseases, including cardiovascular disease, cancer, and metabolic disorder [9]. mTOR exists in two distinct intracellular complexes, including mTORC1 (mTOR Complex 1) and mTORC2 (mTOR Complex 2) [21] (Table 1). A rapamycin-resistant mTOR complex has recently been designated as mTORC3, which is distinct from the conventional mTORC1 and mTORC2 complexes [22]. These complexes differ in composition and function, and they work together to regulate a number of physiological processes within the cells (Figure 1).

### 2.1. mTORC1

#### 2.1.1. Composition of mTORC1

mTORC1 is composed of three core components, namely, mTOR, mLST8 (also known as GβL) [23], and Raptor (regulatory-associated protein of mTOR), along with DEPTOR, an mTOR repressor protein [21,24]. The mTORC1 primarily functions as a key regulator of cell growth by promoting anabolic processes, such as the biosynthesis of proteins, lipids, and organelles, and it also inhibits catabolic processes, e.g., autophagy.

#### 2.1.2. Function of mTORC1

Nutritional and Growth Factor Sensing: mTORC1 senses and binds to growth factors, energy levels, amino acids, and other nutritional and environmental cues to regulate cell growth and metabolism by phosphorylating its downstream substrates. When amino acid levels are low, mTORC1 remains inactive. During energy deprivation or mitochondrial oxidative stress, mTORC1 is inhibited by AMP-activated protein kinase (AMPK), which suppresses all growth-promoting mechanisms and ATP-consuming processes [25,26,27].

Promotion of Anabolism: mTORC1 enhances cell growth by phosphorylating downstream substrates such as S6K (ribosomal protein S6 kinase) and 4E-BP1 (eukaryotic translation initiation factor 4E-binding protein 1), promoting mRNA translation and protein synthesis. In many human cell lines, mTORC1 also facilitates a metabolic shift toward glycolysis and stimulates the pentose phosphate pathway through the activation of HIF-1α (hypoxia-inducible factor-1α) and SREBP1/2 (sterol regulatory element-binding proteins) [28].

Regulation of Autophagy: mTOR plays a central role in regulating autophagy through two major signaling pathways, i.e., the AMPK-mTOR pathway and the PI3K-Akt-mTOR pathway. The classical pathway acts via the PI3K/AKT/mTORC1 signaling axis. Multiple signaling cascades converge on two key protein complexes, namely, the ULK1 (unc-51-like autophagy activating kinase 1) complex and the PI3K lipid kinase complex. The mTORC1 is one of the primary regulators of autophagy by controlling the activity of the ULK1 complex. When mTORC1 is active, it inhibits autophagy by phosphorylating ULK1. However, under various kinds of cellular stress conditions, mTORC1 activity is suppressed, which activates the ULK1 complex and induces autophagy [29].

#### 2.1.3. Regulation of the mTORC1 Signaling Pathway

Upstream regulatory mechanisms:

Activation of PI3K and its downstream effector Akt: (1) Growth factors, e.g., insulin-like growth factor 1 and insulin, can activate the PI3K/Akt signaling pathway. The activation of PI3K and Akt regulates mTORC1 signaling by modulating TSC1/2, thereby reversing the inhibitory effect of the tuberous sclerosis complex (TSC1/2) on mTORC1 [30]. (2) Akt phosphorylates PRAS40, an inhibitor of mTORC1, which causes it to dissociate from Raptor and thus further activates mTORC1. (3) Akt phosphorylates GSK3α/β (glycogen synthase kinase-3α/β), relieving its inhibitory effect on mTORC1 activation. (4) The Wnt signaling pathway, highly conserved during animal development, plays a crucial role in cell proliferation, differentiation, and movement. Although Wnt activity is low in the cardiovascular system of healthy adults, it is reactivated in cardiovascular disease. Wnt signaling inhibits the TSC2-dependent phosphorylation of GSK3, which is independent of β-catenin regulation, thereby activating mTORC1 [31]. (5) Akt phosphorylates Beclin1 (encoded by the BECN1 gene), a Bcl-2-interacting protein with a coiled-coil helical structure. The Akt-dependent phosphorylation of Beclin1 inhibits the activation of the Vps34 complex, the catalytic subunit of the class III PI3K complex [32]. Through these mechanisms, Akt plays a key role in mTORC1 activation.

Role of PTEN: PTEN (phosphatase and tensin homolog) is a well-established tumor suppressor that directly inhibits the PI3K/Akt/mTOR signaling pathway [11].

Inhibition of mTORC1 by AMPK: AMPK (AMP-activated protein kinase) inhibits mTORC1 through two pathways. Firstly, it phosphorylates TSC2, directly inhibiting the activity of Rheb (Ras homolog enriched in the brain) and mTORC1. Secondly, it directly phosphorylates Raptor within mTORC1, inducing a structural change in the complex that inhibits its activity [26,33].

Downstream regulatory mechanisms:

Effect of mTORC1 Activation: Upon activation, mTORC1 phosphorylates 4EBP1 (eukaryotic initiation factor 4E-binding protein 1), which releases eIF4E (eukaryotic translation initiation factor 4E), enhancing the translation of 5′-cap-dependent mRNAs and promoting cellular metabolism and proliferation [34].

Activation of S6K: mTORC1 activation leads to the phosphorylation of S6K (ribosomal protein S6 kinase), which enhances protein synthesis through the activation of eIF4B (eukaryotic translation initiation factor 4B) and promotes cell growth.

Inhibition of Cellular Autophagy by mTORC1: mTORC1 inhibits autophagy by phosphorylating ULK1 (Unc-51-like autophagy activating kinase 1) and autophagy-related protein 13, suppressing the activity of the ULK1 complex [35,36]. Additionally, mTORC1 regulates autophagy by inhibiting the transcription factor EB (TFEB) [36].

### 2.2. mTOR2

#### 2.2.1. Composition of mTORC2 

mTORC2 (mammalian target of rapamycin complex 2) is a finely regulated molecular machine that not only plays a key role in growth factor signaling but also significantly influences the metabolic fate of the cells. The mTORC2 shares several core components with mTORC1, including mTOR, mLST8, mTOR-interacting protein containing the DEP domain, and Tel2-interacting protein 1. However, mTORC2 has specific subunits such as Rictor (rapamycin-insensitive companion of mTOR), Rictor-associated protein 1, and mSIN1 (mammalian stress-activated protein kinase-interacting protein 1), which are essential for its function [37,38,39,40]. These components are critical for substrate binding, determining the subcellular localization of mTORC2, and maintaining the integrity and kinase activity of the complex [41,42,43].

#### 2.2.2. Function of mTORC2

Insulin/PI3K Signaling: mTORC2 functions primarily as an effector of insulin/PI3K signaling, which promotes cell survival and proliferation by phosphorylating the Ser473 site of Akt (protein kinase B).

Cytoskeletal Regulation: mTORC2 regulates cytoskeletal dynamics and cell migration through the phosphorylation of members of the PKC (protein kinase C) family.

Ion Transport and Growth: Additionally, mTORC2 controls ion transport and cell growth by phosphorylating SGK1 (serum- and glucocorticoid-regulated kinase 1).

#### 2.2.3. Regulation of mTORC2 Signaling Pathway

As a key signaling molecule, mTORC2 plays a critical role in cellular signaling, and its regulatory mechanisms are complex. Upstream regulators of mTORC2 include rapamycin, insulin/PI3K, AMPK, and Wnt3A, while its downstream targets involve kinases, e.g., Akt, PKC, and SGK.

##### Upstream Regulatory Mechanisms

Rapamycin: Rapamycin inhibits mTORC1 activity by forming a complex with FKBP12 (FK506-binding protein). Although studies have shown that this complex does not directly affect mTORC2, rapamycin can indirectly influence mTORC2 signaling by interfering with mTOR components.

Insulin/PI3K Signaling: mTORC2 can be activated by growth factors, particularly through the insulin/PI3K signaling pathway. Insulin stimulates PI3K signaling, promoting the binding of mTORC2 to ribosomes, which in turn activates mTORC2 [44]. This process is especially important in melanoma and colon cancer cells. Additionally, during acute energy stress, AMP-activated protein kinase (AMPK) directly activates mTORC2, promoting cell survival [45].

Wnt3A and sNAIL1 (Stinging sensory protein 1): During osteoblast differentiation, Wnt3A induces glycolysis through the activation of mTORC2 [46]. Furthermore, upon phosphorylation by MINK1 (misshapen-like kinase 1), sNAIL1 binds to Rictor and mTORC2, forming a complex that activates mTORC2 to regulate cell adhesion and cancer cell migration [47].

#### 2.2.4. Downstream Regulatory Mechanisms

The AGC kinase family, which includes Akt, PKC, and SGK1 (serine/threonine kinases), represents the primary targets of mTORC2. These kinases are activated or regulated by mTORC2 through the phosphorylation of their hydrophobic motif (HM) and turn motif (TM), which subsequently influences downstream signaling pathways. However, the regulation of AGC kinases by mTORC2 is dynamic. The phosphorylation patterns and downstream effect of AGC kinases may vary depending on the cell type and the physiological or pathological context.

Regulation of Akt: mTORC2 regulates Akt activity by phosphorylating its Ser473 site, a process crucial for normal cardiac function. The persistent activation of the insulin/PI3K/Akt signaling pathway under cardiac overload conditions can lead to myocardial hypoxia and contractile dysfunction [48]. Additionally, mTORC2 regulates the phosphorylation of Akt at the Thr450 site by binding to ribosomes, which increase the stability of nascent Akt and prevents premature ubiquitination [49].

Regulation of PKC: The mTORC2 modulates PKC activity, influencing Ca^2+^ and K^+^ channels on the cell membrane, which in turn affects vascular smooth muscle contraction and cardiac adaptation. PKC is also involved in cell adhesion regulation, and rapamycin inhibits the oxidized LDL-stimulated adhesion molecule expression by interfering with the mTORC2-PKC/c-Fos signaling pathway [39,50,51].

Regulation of SGK1: SGK1, another AGC kinase family member, functions similarly to Akt in signaling. The mTORC2 regulates SGK1, which plays a critical role in autophagy, mitochondrial homeostasis, and nutrient uptake and it is essential for cardiac health and adaptation to pressure overload [52]. Through the activation of SGK1, mTORC2 promotes cardiomyocyte survival, and it is involved in aldosterone-stimulated sodium transport.

Cardiac Adaptation: The inactivation of mTORC2 impairs cardiac adaptation in response to pressure overload, highlighting its importance in maintaining cardiac structure and function.

In summary, mTORC2 finely tunes AGC kinase activity by modulating the phosphorylation of their hydrophobic motifs, which is vital for both cellular physiology and pathological processes.

### 2.3. mTORC3

Harwood et al. identified another rapamycin-insensitive complex known as mTORC3. Notably, mTORC3 lacks the components Raptor, Rictor, and mLST8 that are characteristics of mTORC1 and mTORC2. This complex is capable of phosphorylating targets associated with both mTORC1 and mTORC2. Furthermore, mTORC3 includes ETS transcription factor ETV7, which is crucial for the binding of mTOR and plays an essential role in the assembly of mTORC3 within the cytoplasm [22]. It has been noted that mTORC3 is robustly activated in cancer. The loss of mTORC3 expression in cancer cells has displayed marked sensitivity to rapamycin. Interestingly, the transgenic ETV7 expression further enhances tumor onset and penetrance [53]. Cells expressing ETV7 assemble mTORC3, whose kinase activity induces the increased proliferation and renders cells resistant to rapamycin, indicating that inhibiting mTORC3 may be of therapeutic value for the treatment of disease.

The precise regulation of mTORC1, mTORC2, and mTORC3 is critical for maintaining cellular homeostasis and normal physiological functions. The aberrant activation or inhibition of these complexes has been linked to various cardiac diseases. Therefore, further investigation into the structure, function, and upstream regulation of mTORC1, mTORC2, and mTORC3 is essential for developing new therapeutic strategies.

## 3. The mTOR Pathway and Heart Disease

### 3.1. Cardiac Hypertrophy

Cardiac hypertrophy is typically characterized by cardiomyocyte enlargement and ventricular wall thickening during early overload stress. In the compensatory phase, increased cardiomyocyte volume and mass protect cardiac function. However, sustained stress can lead to irreversible ventricular dilatation, resulting in reduced myocardial contractile function, cardiac decompensation, and heart failure. Myocardial hypertrophy can be categorized as either physiological or pathological. Pathological hypertrophy arises from the prolonged activation of signaling pathways that initially maintain contractility, while it also leads to increased cardiac load and failed compensatory mechanisms. This maladaptation is often accompanied by cardiomyocyte death, fibrosis, mitochondrial dysfunction, metabolic abnormalities, protein damage, fetal gene reprogramming, and inadequate angiogenesis. These processes contribute to poor ventricular remodeling, cardiac dysfunction, and eventual heart failure [54,55,56,57].

Cardiac hypertrophy is closely linked to the dysregulation of autophagy in cardiac cells [58]. In recent years, microRNAs (miRNAs) have been implicated in regulating cardiac disease in cellular and animal models [59]. In cardiac hypertrophy, miRNAs play critical roles by modulating cardiac autophagy. For instance, the miR302-367 cluster was significantly upregulated in h9c2 cells treated with angiotensin II (AngII). This upregulation silences PTEN expression, activates the PI3K/AKT/mTORC1 signaling pathway, reduces cardiac autophagy, and exacerbates cardiac hypertrophy [60]. Similarly, PTEN is a target gene of miR-29a. In a model of aortic arch constriction (TAC)-induced cardiac hypertrophy, the miR-29a level is inversely correlated with PTEN expression. The overexpression of miR-29a inhibits PTEN expression, activates the Akt/mTOR signaling pathway, suppresses autophagy, and ultimately leads to cardiac hypertrophy [61].

Moreover, miR-17-5p is aberrantly expressed in the TAC-induced cardiac hypertrophy model, negatively correlating with the PI3K/AKT/mTORC1 signaling pathway. miR-17-5p inhibits cardiomyocyte autophagy and induces hypertrophy by targeting the mitochondrial fusion protein mitofusin 2 (Mfn2), reducing its expression and activating the PI3K/AKT/mTOR signaling pathway [62].

Additionally, the mTOR signaling pathway is intricately linked to autophagy regulation. Adipose triglyceride lipase (ATGL) expression is significantly reduced in TAC-induced hypertrophied hearts. Mechanistically, ATGL knockout upregulates proteasomal expression and activity, mediates PTEN degradation, and leads to AKT-mTOR signaling activation and autophagy inhibition, which increases cardiac hypertrophy. Interestingly, treatment with proteasomal inhibitors (e.g., epoxomicin) or autophagy activators (e.g., rapamycin) significantly reverses ATGL knockout-mediated cardiac dysfunction [63]. Furthermore, the F-box and WD repeat-containing protein 7 (FBW7) expression is significantly downregulated in hypertrophied hearts. The upregulation of FBW7 increases the expression of autophagy-related proteins LC3B-II and Beclin-1 and activates mTOR-mediated autophagy, which inhibits cellular hypertrophy [64]. These findings highlight the pivotal role of the mTOR signaling pathway in regulating cardiomyocyte autophagy and cardiac hypertrophy. They offer potential new strategies for targeted therapies to treat cardiac hypertrophy and prevent heart failure.

In addition to the regulation of autophagy in cardiac hypertrophy, the growth of cardiomyocyte hypertrophy is influenced by various intracellular signaling factors, such as phosphatidylinositol 3-kinase (PI3K), AKT, and mitogen-activated protein kinase (MAPK), as well as metabolic regulators like AMP-dependent protein kinase (AMPK), NAD^+^-dependent sirtuins, mammalian target of rapamycin (mTOR), and forkhead box subfamily O (FOXO). Histone regulators, including histone deacetylases (HDACs) and demethylases, also contribute to the regulation of hypertrophy [57,65,66,67,68]. The AKT-mTOR signaling pathway plays a crucial role in cardiac hypertrophy, controlling protein synthesis and cardiomyocyte growth through the activation of the S6K1 protein [58]. Carboxypeptidase A4 (CPA4), a novel upstream regulator of mTOR, is significantly upregulated in isoprenaline (ISO)-induced cardiac hypertrophy in mice, promoting hypertrophy by enhancing PI3K-AKT signaling and activating mTOR and downstream S6K1 phosphorylation. Thus, CPA4 contributes to cardiac hypertrophy by activating the PI3K-AKT-mTOR pathway [69].

Similarly, the overexpression of HTR2A, an upstream activator of the PI3K-AKT signaling pathway, increases the phosphorylation levels of PI3K and PDK1 in cardiomyocytes, further activating the AKT-mTOR pathway and driving cardiac hypertrophy [70]. Previous studies have shown that the knockdown of HTR2A in mice also prevents cardiac fibrosis [71]. These studies indicated that a high expression of HTR2A promotes cardiac remodeling, including myocardial fibrosis and hypertrophy, making it a potential therapeutic target for cardiac remodeling.

Additionally, KLK11, highly expressed in hypertrophied cardiomyocytes and the heart, regulates protein synthesis and hypertrophy through mTOR. KLK11 primarily relies on activating key regulators S6K1 and 4EBP1, upstream of the AKT-mTOR signaling pathway, to promote protein synthesis [72]. The AMPK-mTOR pathway is also essential in myocardial hypertrophy. AMPK, a critical regulator of protein synthesis, inhibits mTOR signaling by promoting the activation of the TSC1/TSC2 complex, a direct upstream inhibitor of mTOR. mTOR, in turn, promotes protein synthesis via the activation of ribosomal protein p70 S6K1 [73].

G protein-coupled receptor 39 (GPR39) acts as a negative regulator of AMPK, and it is overexpressed in hypertrophic cells in humans and mice. GPR39 predominantly functions by inhibiting AMPK activation, leading to the enhanced mTOR-S6K1 signaling, which promotes protein synthesis and cardiomyocyte hypertrophic growth [74]. Additionally, the transcription factor EC (TFEC), a basic helix-loop-helix transcription factor, contributes to cardiac hypertrophy by inhibiting AMPK-activated signaling pathways within the mTOR axis [75] (Figure 2).

### 3.2. Cardiomyopathy

Cardiomyopathy is a progressive disease of the myocardium, but it is not caused by abnormal loading conditions or coronary artery disease. Notably, it causes heart failure (HF) and sudden death. Cardiomyopathy can be classified into four types, including dilated cardiomyopathy, hypertrophic cardiomyopathy, restrictive cardiomyopathy, and arrhythmogenic right ventricular dysplasia [76].

#### 3.2.1. Dilated Cardiomyopathy

Dilated cardiomyopathy (DCM) is a non-ischemic form of the disease characterized by the reduced myocardial contractility, ventricular dilation, myocardial fibrosis, and myocyte death [77]. Previous studies have demonstrated that the inactivation of cardiac-specific mTOR, Raptor, and Rictor during both embryonic development and adulthood predisposes individuals to DCM [74]. However, the physiological role of mTOR in postnatal cardiac maturation has not been fully elucidated. In α-MHC-Cre transgenic mice, mTOR inactivation in the early postnatal period has been shown to trigger various pathological changes closely associated with the development of DCM. These changes in the infant heart differ significantly from those observed in adult mice with inducible mTOR or Raptor inactivation [78,79]. Specifically, infant hearts with mTOR conditional knockout (mTORcmKO) exhibit the impaired cardiomyocyte growth, the increased apoptosis, and altered metabolic and oxygen supply patterns. These changes are linked to a significant reduction in the phosphorylation of S6, a substrate of S6K, underscoring the central role of the mTOR signaling pathway in regulating cell growth and survival. Additionally, 4E-BP1 dephosphorylation, coupled with the increased mRNA and total protein levels, further highlights a disruption in translational regulation. This dysregulation, along with the upregulation of Ankrd1 expression, activation of JNK kinase, and accumulation of p53, leads to increased cardiomyocyte apoptosis, defective growth capacity, and enhanced fibrosis, which results in death by the third week in these mice [80].

Recently, a novel Rag GTPase variant (RagC^S75Y) was identified in a patient with syndromic DCM. This de novo missense variant has been shown to cause the hyperactivation of mTORC1 signaling and impaired TFEB function in both in vivo and ex vivo models of RagC^S75Y cardiomyopathy. In neonatal rat ventricular cardiomyocytes (NRVCMs), the RagC variant enhances the increased S6K/S6 phosphorylation while blocking TFEB nuclear translocation, reducing TFEB transcriptional activity. More importantly, the S75Y mutation significantly increases RagC binding to TFEB, which directly affects TFEB’s nuclear translocation [81].

Moreover, mutations in RRAGD, which encodes a small Rag GTPase, have been found to activate the mTOR pathway. The expression of these mutants in HEK293T cells enhances interaction with mTOR, particularly with its regulatory protein Raptor, which promotes S6K1 phosphorylation [82]. The discovery of DCM in a subset of patients further underscores the critical role of Rag GTPases in cardiac function. Previous studies have also shown that the inactivation of Rag GTPases, such as RagA and RagB, can lead to hypertrophic cardiomyopathy in mice. *S75Y* mutation, resulting in the increased mTOR activity, has been described in patients with DCM [83].

Clinical and experimental studies on DCM have consistently shown that mTOR overactivation is a major contributor to the development of DCM, providing new insights into the complex regulatory mechanisms of the mTOR signaling pathway [84].

#### 3.2.2. Hypertrophic Cardiomyopathy

Hypertrophic cardiomyopathy (HCM) is a hereditary cardiac disease characterized by the unexplained left ventricular hypertrophy. mTOR plays a crucial role in regulating protein synthesis, anabolic processes, and autophagy to promote cell growth and survival. The aberrant activation of mTOR in the heart is implicated in the pathogenesis of various cardiomyopathies [85]. The roles of TSC1 and TSC2, major negative regulators of mTOR, have been further investigated in a cardiac-specific Tsc1 knockout mouse model. This study reveals that the mRNA levels of *αB-crystallin* and *HspB2* are significantly elevated in the heart of *Tsc1*-deficient (T1-hKO) mice, along with the hyperphosphorylation of ribosomal protein S6 (p-S6), which activates mTOR and enhances the expression levels of αB-crystallin and HspB2. To explore this mechanism, researchers generated cardiac Tsc1, αB-crystallin, and HspB2 triple-knockout (TKO) mice by crossing T1-hKO mice with αB-crystallin/HspB2 double-knockout (dKO) mice. Compared with T1-hKO mice, tKO mice exhibit the improved cardiac function, reduced heart-to-body weight ratio, and a significant decrease in mortality due to Tsc1 deletion [86].

Furthermore, mutations in the *PTPN11* gene (encoding protein tyrosine phosphatase Shp2) in families with congenital heart disease have been strongly linked to a high prevalence of hypertrophic heart disease. The cardiomyocyte-specific overexpression of the Q510E-Shp2 mutant in neonatal mice induces typical HCM characteristics, including cardiomyocyte hypertrophy, increased heart weight-to-body weight ratio, septal thickening, and abnormal cardiomyocyte arrangement. Signs of interstitial fibrosis are also observed in the hearts of 3-month-old mice. Further analysis reveals that Q510E-Shp2 overexpression leads to the significantly elevated levels of Akt, mTOR, and p70S6K phosphorylation, suggesting that the *Q510E-Shp2* mutation plays a critical role in HCM by activating the mTOR signaling pathway [87].

### 3.3. Myocardial Fibrosis

Myocardial fibrosis is a dynamic process closely associated with ischemia, cardiac hypertrophy, pressure overload, aging, and diabetes. It significantly disrupts the normal structure and function of the myocardium and is a hallmark of advanced stages of several cardiovascular diseases, contributing directly to high mortality and morbidity. The key features of fibrosis include an increase in fibroblasts and excessive deposition of extracellular matrix (ECM) proteins [88]. This overaccumulation of ECM can lead to myocardial stiffness, resulting in diastolic dysfunction and affecting the entire left ventricle, ultimately impairing both dilation and contraction, which can result in heart failure [89].

In recent years, the strong connection between cardiac autophagy and myocardial fibrosis has garnered significant attention. Autophagy, an essential intracellular degradation mechanism, removes the damaged proteins and organelles to maintain cellular homeostasis [90]. During stress overload, autophagy is often inhibited, with increasing protein synthesis and exacerbating myocardial fibrosis while promoting cardiac hypertrophy. However, under stress conditions, the heart can activate autophagy to maintain cardiomyocyte energy supply and counteract ventricular remodeling, highlighting its bidirectional regulatory role [91]. Studies have shown that autophagy is protective by reducing excessive ECM accumulation in the heart [92].

Adenosine monophosphate-activated protein kinase (AMPK), a cellular energy sensor, is a key regulator of cardiomyocyte stress adaptation and plays an important role in activating cardiac autophagy. Recent studies have identified AMPK as an activator of autophagy in cardiac tissues. In mice overexpressing a dominant-negative AMPK, ischemia-induced autophagic vesicle formation was reduced [93]. AMPK initiates autophagy by phosphorylating ULK1 (autophagy initiation kinase), which promotes autophagosome formation, thereby reducing ECM overaccumulation and exerting a protective effect against myocardial fibrosis [94].

In a myocardial fibrosis model, AMPK activity is significantly reduced compared to the normal state, while mTOR activation has been markedly enhanced. This leads to the decreased phosphorylation of ULK1 at the Ser757 site, inhibiting the downstream autophagy pathway and exacerbating myocardial fibrosis. This discovery offers a new perspective in the molecular mechanisms of myocardial fibrosis and highlights potential therapeutic targets [95].

On the other hand, the overstimulation of autophagy in cardiomyocytes can lead to excessive degradation of proteins and organelles [96]. Previous studies have shown that reactive oxygen species (ROS) can exacerbate myocardial ischemia/reperfusion injury by activating autophagy [97]. In some cases, inhibiting autophagic initiation or flux can prevent cell death and alleviate cardiac injury or dysfunction [98,99]. Additional research suggests that autophagy may mediate the transformation of cardiac fibroblasts, implicating it in the onset and progression of myocardial fibrosis [100]. The overexpression of SLP-2, a mitochondrial inner membrane protein abundant in both skeletal muscle and the heart, has been shown to attenuate cardiomyocyte apoptosis, reduce mitochondrial damage, and mitigate myocardial ischemia/reperfusion injury [101]. At 8 weeks, *SLP-2* gene knockout mice have significantly larger areas of cardiomyocytes in the *SLP-2*^−/−^ group than in the WT group, accompanied by a marked increase in cardiac fibroblasts. In the *SLP-2*^−/−^ Dox group, the expression levels of autophagy-related proteins (LC3BII, Beclin1, and ATG5) are notably higher compared to the WT group. The ratios of phos-PI3K/PI3K, phos-Akt/Akt, and phos-mTOR/mTOR proteins are significantly lower in the Dox group than in the saline group, with SLP-2 deficiency further reducing these ratios. This suggests that the absence of SLP-2 suppresses the PI3K-Akt-mTOR signaling pathway, leading to excessive autophagy activation [102]. Cardiac fibrosis is a complex pathological process regulated by multiple factors. Several cell cycle genes have been identified as key regulators of cardiac fibroblast proliferation and migration, contributing to cardiac fibrosis [103]. BMI1, a transcriptional repressor, modulates the PTEN-PI3K/Akt-mTOR signaling pathway. By inhibiting the expression of phosphatase and tensin homolog (PTEN), BMI1 activates the phosphatidylinositol 3-kinase (PI3K)/Akt pathway, facilitating the proliferation and migration of various tumor cells [104,105]. mTOR, a well-known downstream factor of the PI3K/Akt pathway, plays a crucial role in fibrosis [106]. In ischemia-induced myocardial fibrosis, BMI1 expression is upregulated. The overexpression of BMI1 suppresses PTEN, enhances PI3K expression, and increases the phosphorylation of Akt and mTOR, which promotes the proliferation and migration of cardiac fibroblasts. This effect is reversible with PI3K/mTOR inhibitors, indicating that the PTEN-PI3K/Akt-mTOR pathway is a potential mechanism through which BMI1 exacerbates myocardial fibrosis and worsens cardiac function after myocardial infarction [107].

### 3.4. Myocardial Ischemia–Reperfusion Injury

Myocardial ischemia/reperfusion injury (MIRI) is a complex pathological process that occurs when coronary arteries are partially or completely blocked, leading to a progressive deterioration of myocardial ischemia. This process results in serious changes in the myocardium, including electrophysiological dysfunction, ultrastructural damage, and imbalances in energy metabolism. In severe cases, it can even induce life-threatening arrhythmias [108]. The rapid restoration of blood flow, or reperfusion, is crucial to saving the ischemic myocardium. However, rapid reperfusion can trigger a series of adverse events, such as the exacerbation of endoplasmic reticulum stress, calcium ion (Ca^2+^) overload, and excessive production of ROS, collectively known as ischemia–reperfusion (I/R) injury [109,110].

Studies using a genetic mouse model have shown that overexpression of mTOR significantly reduces mortality following I/R and effectively prevents unfavorable left ventricular remodeling, thus protecting cardiac function. In vitro cardiac perfusion experiments have further demonstrated that mTOR inhibits cardiomyocyte necrosis, reduces inflammatory responses, and promotes the recovery of cardiac function [111]. Despite these findings, the precise mechanisms by which mTOR affects cardiac function during ischemia–reperfusion remain unclear.

Interestingly, studies utilizing mTOR knockout models have shown that although the deletion of cardiac mTOR initially seems to promote recovery after I/R, during reperfusion, the mTOR knockout (CKO) heart exhibits irregular contractility, a phenomenon not observed in the control (Con) heart. At the cellular level, the total calcium content of the sarcoplasmic reticulum (SR) is significantly lower in the *mTOR* CKO heart compared to cardiomyocytes from the control heart. This reduction in SR calcium may be linked to the impaired function in *mTOR* CKO cardiomyocytes following reperfusion, highlighting the need for further exploration of calcium dysregulation mechanisms after I/R [112].

The mTOR signaling pathway and its related components play a central role in cardiac hypertrophy, ischemia, autophagy, and cardiomyopathy, which reflects potential therapeutic targets. However, heart disease as a dynamic process is associated with multiple factors in which autophagy plays a bidirectional regulatory role, either in protecting the heart by removing harmful substances or potentially damaging myocardial structure and function due to overactivation. These studies not only deepen our understanding of the mechanisms of heart disease but also provide a theoretical basis for the development of new therapeutic approaches.

## 4. mTOR Pathway and Heart Disease Treatment

mTOR is a crucial intracellular protein that plays a pivotal role in regulating fundamental cellular processes, positioning the modulation of the mTOR signaling pathway as a key therapeutic target for cardiovascular diseases [18,113,114]. Within the broader context of mTOR pathway regulation, several approaches hold promise for clinical applications and research, including pharmacological interventions, cardiomyocyte transplantation, gene therapy, and molecular targeting strategies for cardiac disease treatment [115].

Pharmacological intervention is often the primary strategy, with rapamycin and its derivatives, collectively known as first-generation mTOR allosteric inhibitors, being the most used drugs. Since its discovery, rapamycin has been widely applied in various fields, and recent studies have shown progress in its use for cardiac disease research [116]. Additionally, derivatives such as metformin, Sirolimus, rapamycin, and Everolimus have been employed in treating cardiomyopathy, heart failure, and other cardiovascular conditions [117,118,119] (Table 2).

Despite extensive research on first-generation mTOR inhibitors (Rapalogs), several limitations have been identified, such as large molecular size, complex structure, limited modification sites, and difficulties in synthesis. These drawbacks have prompted the development of second-generation mTOR inhibitors, known as ATP-competitive inhibitors. Molecules such as AZD2014, which have smaller molecular weights and simplified structures, can overcome the limitations of first-generation inhibitors [120]. These inhibitors work by selectively targeting the active kinase site of mTOR, thus obtaining their classification as ATP-competitive inhibitors.

The development of third-generation mTOR inhibitors marks a significant advancement in the field. These inhibitors use a linker to combine rapamycin with ATP-competitive inhibitors, effectively overcoming resistance in mutant forms of *mTOR*. However, both second- and third-generation inhibitors are still in the experimental stages, and further clinical development is required before they can be widely adopted (Table 3).

### 4.1. Diabetic Cardiomyopathy

Diabetic cardiomyopathy (DCM), an early complication of diabetes, is characterized by myocardial fibrosis, ventricular stiffness, and cardiac enlargement, often manifesting without overt clinical symptoms. DCM is a serious diabetic complication independent of coronary artery disease, hypertension, heart failure, or other cardiac conditions. Research on diabetic heart disease has shown that ALDH2 regulates mitochondrial fusion and fission by inhibiting the opening of the mitochondrial permeability transition pore (mitoPTP) and activating the PI3K/AKT/mTOR signaling pathway, thereby mitigating ischemia–reperfusion injury in DCM [121]. Adipokine Neuregulin-4 (Nrg4) protects against metabolic disorder and insulin resistance [122]. Studies have demonstrated that Nrg4 activates autophagy via the AMPK/mTOR signaling pathway, upregulating autophagy and providing cardioprotection by significantly reducing myocardial fibrosis. Diabetes has been known to reduce AMPK phosphorylation and increase mTOR to inhibit autophagy. Nrg4 effectively counteracts these diabetic effects for protection [123].

**Table 3 biomedicines-13-00397-t003:** Advantages and disadvantages of three generations of mTOR inhibitors.

Category	mTOR Inhibitor	Advantages	Disadvantages	Refs.
Rapalog	Rapamycin	Wide effective dose range; low toxicity; no patent; relatively low price; persistent effect on cardiac function even after discontinuation; approved by the FDA for sale.	There may be side effects, such as oral ulcers, hyperglycemia, and hyperlipidemia, but they can subside after discontinuing the medication; immunosuppressive effect increases the risk of infection; difficult to dissolve in water; low bioavailability; large molecular weight; complex structure; limited restriction sites.	[124,125,126,127]
Rapalog	Temsirolimus	FDA-approved for sale; good anti-cancer activity in various types of malignant tumors; low dosage; strong efficacy.	Possible side effects include hypercholesterolemia and hyperglycemia, but they can be controlled through diet or medication. Its efficacy and safety are severely limited by defects in solubility and bioavailability, making it poorly soluble in water.	[128]
Rapalog	Everolimus	FDA-approved for sale; has been used for various neuroendocrine tumors of the digestive tract, lungs, and pancreas; not only does it inhibit the growth and proliferation of APT cells, but it also enhances their sensitivity to radiotherapy and chemotherapy.	It can cause adverse reactions, commonly including hyperlipidemia, oral ulcers, and hyperglycemia. Large molecular weight, complex structure, difficult to synthesize, limited binding sites.	[129,130,131]
Rapalog	Ridaforolimus	It has good overall tolerance, and water solubility, affinity, and stability have been improved significantly compared to rapamycin.	It may cause side effects, such as rashes, oral ulcers, and high blood sugar. It has a high molecular weight and a complex structure and is difficult to synthesize.	[132,133]
ATP-competitive	AZD8055	Exhibits excellent selectivity for all class I phosphoinositide 3-kinase (PI3K) isoforms and other members of the PI3K-like kinase family; in inhibiting erythroid differentiation and promoting mitochondrial clearance, it is more effective than rapamycin. Small molecular weight; simplified structure.	Hepatotoxicity is strong, limiting clinical application.	[133,134]
ATP-competitive	Torkinib	It can inhibit mTORC1 and mTORC2 and has a neuroprotective effect. Small molecular weight; simplified structure.	Hepatotoxic.	[135,136]
ATP-competitive	Torin1	It can selectively inhibit mTORC1 and mTORC2; small molecular weight; simplified structure.	Poor stability, low oral bioavailability, and a very short half-life hinder further in vivo studies.	[137,138]
ATP-competitive	Torin2	Torin2 is derived from the chemical simplification of Torin1, with better stability and a longer half-life.	The inhibitory effect is not as strong as that of Torin1.	[138]
Rapalink	Palomid529	Its characteristic lies in its lack of affinity for ABCB1/ABCG2 (two drug efflux transporters) and its good brain penetration. It can enhance the effects of radiotherapy.	Poor solubility, low oral bioavailability of micronized form.	[139,140]
Rapalink	Rapalink1	Low dose, high efficacy. Compared to rapamycin, Rapalink1 has higher specificity for mTORC1. Good permeability to the blood–brain barrier.	It does not inhibit the substrate of mTORC2.	[141,142]

ABCB1/ABCG2—two drug efflux transporters.

Metformin, a commonly used anti-diabetic drug, also acts via the mTOR pathway to prevent diabetic heart disease. It reduces cardiac fibrosis by inhibiting the iNOS/mTOR/TIMP-1 signaling cascade and activating AMP-activated protein kinase (AMPK), a tissue-protective enzyme [143]. In addition to metformin, sodium–glucose co-transporter 2 (SGLT2) inhibitors have emerged as promising therapeutic agents for DCM, demonstrating cardiovascular protective effects, particularly with the use of SGLT2 inhibitors (SGLT2is) [144]. Although the heart lacks SGLT2 channels, selective SGLT2 targets the long-activated sodium–hydrogen exchanger (NHE), reducing intracellular sodium levels. This action increases intracellular pH, making it more alkaline, which activates AMPK and inhibits mTORC1, thereby providing cardioprotection. Additionally, SGLT2 inhibitors simulate energy-deprived conditions, inducing Sestrin2 expression, which activates AMPK and SIRT1, leading to mTORC1 inactivation and cardiovascular protection.

Furthermore, quercetin (QUE) has been found to regulate autophagy and apoptosis following myocardial injury in diabetic rats. Quercetin modulates autophagy and inhibits apoptosis through the AMPK/mTOR signaling pathway, thereby reducing myocardial injury and improving cardiac function in diabetes [145].

### 4.2. Heart Failure

Heart failure (HF) is the syndrome resulting from ventricular dysfunction and is associated with high morbidity and mortality. Left ventricular failure presents symptoms such as shortness of breath and fatigue, while right ventricular failure leads to peripheral tissue and fluid retention. Both ventricles may be affected simultaneously or individually, with diagnosis primarily based on clinical presentation. Numerous studies have identified effective treatment for chronic heart failure, many of which target the inhibition of the Akt/mTORC1 pathway to enhance autophagic flux [146]. Myocardial infarction (MI) is a major cause of both HF and mortality. The gene-edited Klotho protein, an anti-aging factor related to cardiovascular disease, has been shown to reduce cardiac fibrosis, inhibit inflammatory cytokines, and improve the pathological state of MI. Its cardioprotective effect in HF is linked to the mTOR pathway, since it induces cardiomyocyte autophagy by inhibiting the PI3K/AKT/mTOR signaling cascades [147].

In addition to endogenous protein functions, traditional Chinese medicine (TCM) plays a distinct role in preventing and treating cardiovascular diseases, with safety and efficacy. TCM research includes compound herbal formulations and single-herb therapies for HF prevention. For example, Yangxin Kang, a compound herbal formula, prevents post-MI heart failure and cardiac remodeling by inhibiting autophagy through the modulation of the AMPK/mTOR pathway. Yangxin Kang reduces the phos-AMPK level while increasing phos-mTOR expression, significantly suppressing AMPK/mTOR signaling [148]. In single-herb studies, gardenoside has been shown to prevent HF by activating autophagy via the AMPKα/mTOR signaling pathway and inhibiting endoplasmic reticulum stress [149]. Chronic HF progression also disrupts and reduces autophagy. Moxibustion, a TCM therapy, alleviates chronic HF by upregulating mTOR expression and inhibiting autophagy [150].

Additionally, Dioscin has been reported to treat MI by promoting angiogenesis [151]. Furthermore, Dioscin improves doxorubicin (Dox)-induced heart failure by inhibiting autophagy and apoptosis through the regulation of the PDK1-mediated Akt/mTOR signaling pathway [152].

### 4.3. Cardiac Hypertrophy

Cardiac hypertrophy is a powerful compensatory mechanism; however, it has limitations. If the underlying cause remains unresolved over time, the hypertrophied myocardium will eventually lose its ability to maintain normal function, ultimately progressing to heart failure. Chronic heart failure typically evolves gradually from compensatory myocardial hypertrophy. Yiqi Fumai lyophilized injection (YQFM) has been shown to protect against myocardial hypertrophy and arrhythmia [153]. YQFM exerts its protective effect by activating the PI3K/AKT/mTOR signaling pathway, which reduces apoptosis and regulates autophagy in hypertrophic myocardium.

Among herbal medicines, Berberine (an active component of Coptidis rhizoma) has been demonstrated to possess broad cardiovascular protective effect. The Akt-mTOR axis plays a crucial role in regulating eccentric hypertrophy, with mTOR activity determining the rate of its progression. Berberine has been shown to inhibit mTOR phosphorylation, block signal transduction, and inactivate downstream proteins such as p70 S6K and 4EBP1, thereby preventing cardiac hypertrophy [154]. Beyond its role in autophagy regulation, mTOR is also involved in mitochondrial oxidative and reductive processes. Recent studies have identified Sirtuin 3 (SIRT3) as a conserved NAD-dependent deacetylase in mitochondria. The 2-APQC, a small-molecule activator of SIRT3, can inhibit the AKT/mTOR/p70S6K signaling pathway via SIRT3 activation to improve the ISO-induced cardiac hypertrophy and myocardial fibrosis [155].

Resveratrol, a polyphenolic compound, is widely used to improve systemic pathological conditions. Research has shown that CIH treatment significantly enhances cardiac function and reduces cardiac hypertrophy, oxidative stress, and apoptosis. It was also demonstrated that after CIH stimulation, resveratrol mediated the blockade of NLRP3 inflammation through the activation of AMP-activated protein kinase (AMPK) and could inhibit mTOR/TTP/NLRP3 mRNA signaling [156].

### 4.4. Myocardial Ischemia–Reperfusion Injury

Myocardial ischemia–reperfusion injury (MIRI) refers to the aggravated damage that occurs in ischemic myocardium following the restoration of blood flow, which is more severe than ischemia alone. The optimal intervention for acute myocardial infarction (AMI) is early myocardial reperfusion through revascularization. However, MIRI can worsen the prognosis after revascularization and increase long-term mortality [157]. Myocardial cell stress and reperfusion may result in mitochondrial dysfunction, leading to an energy crisis, oxidative stress, and inflammation, which can induce autophagy [158]. Changes in energy homeostasis affect mTOR, a critical regulatory protein involved in multiple therapeutic mechanisms. Many drugs targeting MIRI exert their effect by modulating the mTOR pathway. For instance, the traditional Chinese medicine Su Xiao Jiu Xin Wan (SJP) has been demonstrated notable efficacy in treating MIRI. Its molecular mechanism involves the ALKBH5/GSK3β/mTOR pathway, through which SJP provides cardioprotection by alleviating myocardial reperfusion injury [159]. Another effective drug, Tongguan Capsule (TGC), alleviates MIRI and protects the heart by upregulating autophagy-related proteins via the mTOR pathway [160].

MIRI often leads to cardiac dysfunction, and studies have shown that estrogen has a beneficial impact on MIRI. 16α-OHE1, a novel estrogen metabolite, reduces MIRI and mitigates left-ventricular systolic dysfunction by regulating the AMPK/mTOR pathway and activating autophagy [161]. Shifting the focus to herbal medicine, L-Borneol 7-*O*-[*β*-d-apiofuranosyl-(1→6)]-*β*-d-glucopyranoside (LBAG), a natural derivative of Ophiopogon japonicus, may limit MIRI-induced apoptosis and autophagy by stimulating the PI3K/mTOR pathway in rat hearts [162]. This explains the cardioprotective effect of Ophiopogon japonicus. Additionally, recent research on Panax notoginseng has identified its mechanism in inhibiting excessive autophagy: Panax notoginseng saponins (PNS) suppress myocardial autophagy by inhibiting the PI3K/Akt/mTOR signaling pathway [163].

In summary, the mTOR signaling pathway plays key roles in various cardiac diseases and provides a theoretical basis for developing new therapeutic approaches. Modulating this pathway can achieve effective intervention in diabetic cardiomyopathy, heart failure, cardiac hypertrophy, and myocardial ischemia/reperfusion injury (Table 4).

## 5. Conclusions

In this review, we highlight the dual roles of mTOR in heart disease, where it can either support cardiac function or contribute to disease progression, depending on the context. Given its involvement in cardiac hypertrophy, ischemia, autophagy, and cardiomyopathy, mTOR represents a critical target for potential therapeutic interventions aimed at treating or preventing heart disease. Furthermore, mTOR inhibitors, such as rapamycin, offer a promising therapeutic approach for both the treatment and prevention of heart disease. In conclusion, the mTOR signaling pathway remains a highly promising target for therapeutic strategies in heart disease.

## Figures and Tables

**Figure 1 biomedicines-13-00397-f001:**
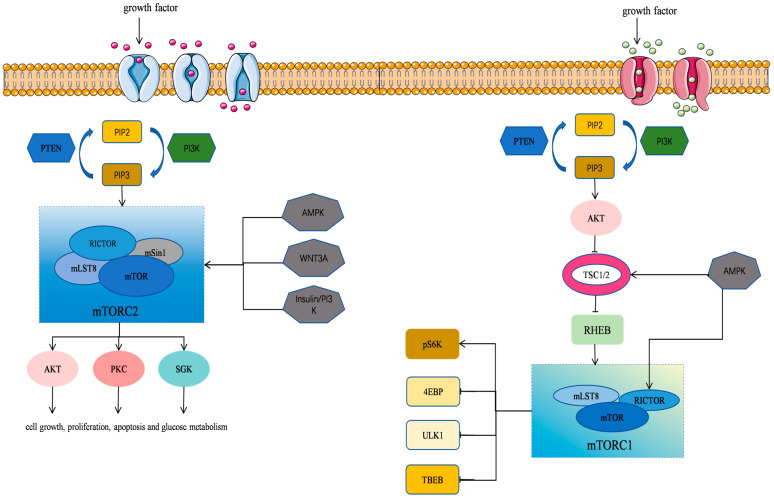
mTOR signaling pathway in cells. The mTOR’s structure and function as well as the regulatory mechanisms of its associated signaling pathways are illustrated.

**Figure 2 biomedicines-13-00397-f002:**
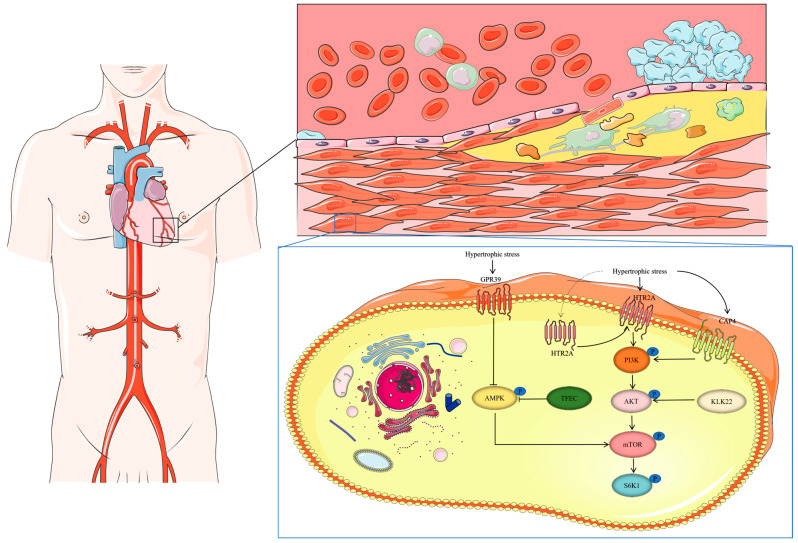
The function of GPR39, TFEC, KLK22, CAP4, and HRT2A during myocardial hypertrophy. Under hypertrophic stress, the KLK22, CAP4, and HTR2A expression is upregulated, leading to the activation of PI3K. Activated PI3K promotes AKT activation in a PDK1-dependent manner. GPR39 and TFEC overexpression leads to AMPK inhibition. The activation of mTOR and S6K1 signaling results in the increased protein synthesis and hypertrophic growth of cardiomyocytes.

**Table 1 biomedicines-13-00397-t001:** Composition, function, and upstream and downstream regulatory mechanisms of mTORC1 and mTORC2.

	mTORC1	mTORC2
Composition	mTOR, mLST8 (also known as GβL), Raptor, and DEPTOR	mTOR, mLST8, DEPTOR, Tel2-interacting protein 1, and specific subunits such as Rictor, Rictor-associated protein 1, and mSIN1
Function	Nutritional and growth factor sensing, promotion of anabolism, and regulation of autophagy	Promoting cell survival and proliferation by insulin/PI3K signaling, cytoskeletal regulation, and ion transport and growth
Upstream regulatory mechanisms	Activation of PI3K and its downstream effector Akt, inhibition of mTORC1 by AMPK and PTEN	Inhibition of mTORC2 by rapamycin, activation of mTORC2 by Wnt3A, sNAIL1, growth factors (Insulin/PI3K Signaling), and acute energy stress (AMPK)
Downstream regulatory mechanisms	Upon activation, mTORC1 phosphorylates 4EBP1 (which releases eIF4E), activates S6K, and regulates autophagy by inhibiting the transcription factor EB and phosphorylating ULK1 and autophagy-related protein 13	Regulation of Akt, PKC, and SGK1(through the phosphorylation of their hydrophobic motif (HM) and turn motif (TM))

DEPTOR—an mTOR repressor protein; Rictor—rapamycin-insensitive companion of mTOR; Raptor—regulatory-associated protein of mTOR; mSIN1—mammalian stress-activated protein kinase-interacting protein 1; AMPK—AMP-activated protein kinase; PTEN—phosphatase and tensin homolog; S6K—ribosomal protein S6 kinase; 4EBP1—eukaryotic translation initiation factor 4E-binding protein 1; eIF4B—eukaryotic translation initiation factor 4B; ULK1—unc-51-like autophagy activating kinase 1.

**Table 2 biomedicines-13-00397-t002:** Application of mTOR inhibitors in clinical trials.

mTOR Inhibitor	Cardiopathy	Phase	Statues	Trail Identifier
Metformin	Heart failure	II	Terminated	NCT03331861
Everolimus	Heart infarction	IV	complete	NCT01347554
Everolimus	Coronary heart disease	III	complete	NCT00180479
Sirolimus	Ischemic heart disease	III	complete	NCT00476957
Rapamycin	Heart failure	I	complete	NCT04996719
Metformin	Ischemic heart disease	IV	complete	NCT01438723

**Table 4 biomedicines-13-00397-t004:** Drugs targeting mTOR to treat heart disease.

Cardiopathy	Signal Pathways	Drugs	References
Diabetic cardiomyopathy	PI3K/Akt/mTOR	ALDH2	[121]
Diabetic cardiomyopathy	AMPK/mTOR	Neuregulin-4; SGLT2is; Quercetin	[123,144,145]
Diabetic cardiomyopathy	INOS/mTOR/TIMP-1	Metformin	[143]
Heart failure	PI3K/Akt/mTOR	Dioscin	[147,152]
Heart failure	AMPK/mTOR	Effect of Yangxinkang Tablets; geniposide	[148,149]
Myocardial Hypertrophy	PI3K/Akt/mTOR	YQFM; Resveratrol	[151]
Myocardial Hypertrophy	AMPK/mTOR	Resveratrol	[154]
Myocardial Hypertrophy	Akt/mTOR/p70S6K	Berberine; 2-APQC	[154,155]
Myocardial ischemia/reperfusion injury	ALKBH5/GSK3β/mTOR	SJP	[159]
Myocardial ischemia/reperfusion injury	AMPK/mTOR	16α-OHE1	[161]
Myocardial ischemia/reperfusion injury	PI3K/Akt/mTOR	LBAG; PNS; Radix Ophiopogonis; TGC	[160,162,163]

ALDH2—Aldehyde dehydrogenase 2; SGLT2is—sodium–glucose cotransporter-2 inhibitors; YQFM—Yiqi Fumai lyophilized injection; APQC—small-molecule activator of Sirtuin-3 (SIRT3); SJP—Chinese patent medicine Suxiao Jiuxin Pill; LBAG—L-Borneol 7-*O*-[*β*-d-apiofuranosyl-(1→6)]-*β*-d-glucopyranoside; PNS—Panax ginseng total saponin; TGC—Tongguan capsule; 16α-OHE1—novel estrogen metabolite.

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
