# Peer review of "The mTOR Signaling Pathway: Key Regulator and Therapeutic Target for Heart Disease"

_biomedicines, 2025, doi:10.3390/biomedicines13020397_

Round 1

Reviewer 1 Report

Comments and Suggestions for Authors

Dr. Yang and the group have authored a timely review article exploring the role of the mTOR signaling pathway in heart disease. This well-written manuscript highlights the latest findings in the field and delves into the therapeutic implications of mTOR signaling, enhancing the translational relevance of the review. However, a few key points need to be addressed before it is ready for acceptance, as outlined below:

1. It is a well-established fact that a reciprocal feedback mechanism between AMPK and the mTOR signaling pathway holds significant therapeutic potential. This mechanism suggests that combining the AMPK activator metformin with the mTOR inhibitor rapamycin could have therapeutic benefits in various diseases (PMID: 26323019, PMID: 32276645). The authors should include a discussion on this aspect, incorporating the mentioned references and focusing on the relevance of this combination specifically in the context of heart disease.

2. The authors should include a table summarizing ongoing clinical trials targeting mTOR signaling in heart diseases.

3. The authors should consider revising the title to make it more professional and engaging for readers. As an alternative, they can think something like this "The mTOR Signaling Pathway: Key Regulator and Therapeutic Target in Heart Disease". 

Author Response

We feel great thanks for your professional review work on our manuscript. In the following pages are our point-by-point responses to each of your comments. Thank you again for your time, effort, and very helpful comments, which have helped us to improve and perfect our manuscript.

  1. It is a well-established fact that a reciprocal feedback mechanism between AMPK and the mTOR signaling pathway holds significant therapeutic potential. This mechanism suggests that combining the AMPK activator metformin with the mTOR inhibitor rapamycin could have therapeutic benefits in various diseases (PMID: 26323019, PMID: 32276645). The authors should include a discussion on this aspect, incorporating the mentioned references and focusing on the relevance of this combination specifically in the context of heart disease.

Response:Thank you very much for your suggestion.We have carefully read the literature you provided and other relevant literature, due to the combination of AMPK activator metformin and mTOR inhibitor rapamycin hardly mentioned heart disease, we cannot cite.We mention in lines 547 to 558 of the revised manuscript that metformin reduces cardiac fibrosis to prevent diabetic heart disease by inhibiting the iN-OS/mTOR/TIMP-1 axis and activating AMP-activated protein kinase (AMPK).

  1. The authors should include a table summarizing ongoing clinical trials targeting mTOR signaling in heart diseases.

Response: Thanks for your advice.Due to we discuss the treatment of drugs through mTOR targets in heart disease, we ignore the clinical trials targeting mTOR signaling in heart diseases.So we have inserted table 2 in the revised manuscript to summarize application of mTOR inhibitors in clinical trials.

  1. The authors should consider revising the title to make it more professional and engaging for readers. As an alternative, they can think something like this "The mTOR Signaling Pathway: Key Regulator and Therapeutic Target in Heart Disease". 

Response: Thank you very much for your suggestion.We have replaced the headline"the role of mTOR signaling pathway in heart disease" with "The mTOR Signaling Pathway: Key Regulator and Therapeutic Target in Heart Disease".

We appreciate your warm work earnestly and hope that our corrections will be recognized. Once again, thank you very much for your comments and suggestions.

Reviewer 2 Report

Comments and Suggestions for Authors

Please find enclosed the review report

Author Response

We feel great thanks for your professional review work on our manuscript. In the following pages are our point-by-point responses to each of your comments. Thank you again for your time, effort, and very helpful comments, which have helped us to improve and perfect our manuscript.

1 . Please take care of the typos in the manuscript. For example, the title should be with "T" and not "t". To whom of the authors belong the affiliation Keywords: take care of the semicolons English should be revised throughout the manuscript (singular, plural forms, articles in some places are missing) All references throughout the manuscript body are wrongly placed (there is a lack of space before the reference number in square brackets). In some places, the references are lacking (e.g., line 27 after burden on society). Additionally, the References at the end are not correctly formatted.

Response:Thank you very much for pointing out our shortcomings. We are sorry for the mistakes. The whole manuscript grammar has been checked and revised, marked in yellow. the position of references and the format of endings have been revised according to the template, and the author's affiliation  have been changed. A new reference has been added in line 27.

2.(1)Chapter 2: The mTOR pathway and heart disease

The authors did not mention putative mTORC3 – even if the Authors do not describe it, it is worth mentioning.

Response:Thank you very much for your suggestion.We have added part 2.3 to the manuscript.

(2)The chapters 2.1. and 2.2. Both chapters are rich in details, which makes them hard to follow. It would be better to summarize both chapters in one table, e.g., composition, primary function,upstream regulators,downstream targets, rapamycin sensitivity, autophagy regulation(direct/indirect?), and role in hypertrophy. Also, key differences and similarities should be summarized to give the readers key messages that the authors want to convey in this chapter.

Response:Thanks for your advice. We have inserted table 1 in the revised manuscript to summarize composition, function, upstream and downstream regulatory mechanisms of mTORC1 and mTORC2.

3.The chapter 3.1. The chapter also should be summarized at the end, giving the readers a take-home message.

Response:We are very appreciated for your comments and suggestions. We added a summary at the end of Chapter 3.3.

4.The chapter 4

It is the most important and interesting chapter for the readers. The chapter focuses onthe therapeutic implications of the mTOR pathway in treating heart conditions. You highlighted three generations of used or potential use mTOR inhibitors. Also, this chapter explores the applications of mTOR pathway modulation in different heart diseases.Adding diagrams/figures or tables showing (1) mTOR generation-dependent inhibitors, including effectiveness and limitations, and (2) mTOR pathway modulation in different heart diseases would be more beneficial. The second does not have to be a figure; it could be a table summarizing the heart diseases, therapeutic approaches mentioned, and mTOR pathway involvement (with references).

Response: Thank you very much for your review and suggestions. Due to we discuss the treatment of drugs through mTOR targets in heart disease, we ignore the effectiveness and limitations of third-generation mTOR inhibitors.So we have inserted table 3 in the revised manuscript to summarize advantages and disadvantages of three generations mTOR inhibitors,we summarize at the end of Chapter 4 and insert table 4 to summarize Drugs targeting mTOR to treat heart disease.

We appreciate your warm work earnestly and hope that our corrections will be recognized. Once again, thank you very much for your comments and suggestions.

Round 2

Reviewer 1 Report

Comments and Suggestions for Authors

accepted

Author Response

We sincerely appreciate your recognition and thank you again for your comments and suggestions.

Reviewer 2 Report

Comments and Suggestions for Authors

The authors responded to the concerns raised in my review. Due to the introduction of new tables, which I believe improve the article's comprehensibility for readers, it would be advisable to include footnotes (if relevant) in the tables that explain all the abbreviations, making the tables self-explanatory.
Additionally, Table 1 is mentioned in the text on page 2 (line 81), while the table itself only appears on page 6; Table 3 is cited in the text in line 536, after it has already appeared in the manuscript in line 522

Author Response

The authors responded to the concerns raised in my review. Due to the introduction of new tables, which I believe improve the article's comprehensibility for readers, it would be advisable to include footnotes (if relevant) in the tables that explain all the abbreviations, making the tables self-explanatory.
Additionally, Table 1 is mentioned in the text on page 2 (line 81), while the table itself only appears on page 6; Table 3 is cited in the text in line 536, after it has already appeared in the manuscript in line 522.

Response:Thanks for your comment. We are sorry for the mistakes. We have mainly added footnotes under tables 1 and 4, marked in red. Tables 1 and 3 have been adjusted.                                             

We appreciate your warm work earnestly and hope that our corrections will be recognized. Once again, thank you very much for your comments and suggestions.